# Research Progress of Small Molecule VEGFR/c-Met Inhibitors as Anticancer Agents (2016–Present)

**DOI:** 10.3390/molecules25112666

**Published:** 2020-06-08

**Authors:** Qian Zhang, Pengwu Zheng, Wufu Zhu

**Affiliations:** Jiangxi Provincial Key Laboratory of Drug Design and Evaluation, School of Pharmacy, Jiangxi Science & Technology Normal University, Nanchang 330013, China; 15797966937@163.com (Q.Z.); zhengpw@126.com (P.Z.)

**Keywords:** dual VEGFR/c-Met inhibitors, anticancer agents, research progress

## Abstract

Vascular endothelial growth factor receptor 2 (VEGFR-2) binds to VEGFR-A, VEGFR-C and VEGFR-D and participates in the formation of tumor blood vessels, mediates the proliferation of endothelial cells, enhances microvascular permeability, and blocks apoptosis. Blocking or downregulating the signal transduction of VEGFR is the main way to discover new drugs for many human angiogenesis-dependent malignancies. Mesenchymal epithelial transfer factor tyrosine kinase (c-Met) is a high affinity receptor for hepatocyte growth factor (HGF). Abnormal c-Met signaling plays an important role in the formation, invasion and metastasis of human tumors. Therefore, the HGF/c-Met signaling pathway has become a significant target for cancer treatment. Related studies have shown that the conduction of the VEGFR and c-Met signaling pathways has a synergistic effect in inducing angiogenesis and inhibiting tumor growth. In recent years, multi-target small molecule inhibitors have become a research hotspot, among which the research of VEGFR and c-Met dual-target small molecule inhibitors has become more and more extensive. In this review, we comprehensively summarize the chemical structures and biological characteristics of novel VEGFR/c-Met dual-target small-molecule inhibitors in the past five years.

## 1. Introduction

The occurrence of tumor is a multi-stage and complex process which seriously endangers human life and health [1]. Tumor metastasis, growth and survival depend on cell differentiation, proliferation, angiogenesis and apoptosis, which are regulated by a variety of signal transduction pathways and protein kinases [2,3]. At present, cancer therapy that interferes with a single biomolecule or pathway has been successfully applied [4]. However, the problem of drug resistance often arises in the research of single target drugs and combination drugs [4,5]. It is found that multi-target drugs may overcome drug resistance and achieve higher efficacy than single target drugs, which makes the molecules of multi-target drugs widely studied [4,5,6].

The vascular endothelial growth factor (VEGF) pathway is one of the most essential active regulators of angiogenesis. It can promote the proliferation and migration of vascular endothelial cells and induce the formation of blood vessels [7,8]. There are three main vascular endothelial growth factor receptors (VEGFR-1, VEGFR-2 and VEGFR-3), which are the key intermediate products of tumor angiogenesis and new blood vessels and provide nutrition and oxygen for tumor growth [9]. The binding of VEGF with the receptor leads to the homologous or heterogenous dimerization of the receptor, phosphorylation of the kinase area in the cell, activation of many major signaling pathways, and production of many physiological effects [10]. Vascular endothelial growth factor receptor-2 (VEGFR-2) is the main effector of VEGF/VEGFR signal transduction in promoting tumor angiogenesis. It is expressed on the surface of blood vessels and plays a key role in tumor angiogenesis [11,12]. The phosphorylation of VEGFR-2 activates the Raf-1/MAPK/ERK signaling pathway, which will eventually lead to angiogenesis, enhanced vascular permeability, tumor proliferation and tumor migration [13]. Therefore, inhibition of the VEGFR-2/VEGF signaling pathway is considered to be one of the most eventful and valuable pathways in the development of tumor chemotherapy [10]. At present, a number of VEGFR-2 inhibitors approved by the Food and Drug Administration (FDA) are used as chemotherapy drugs in clinical cancer treatment [14]. However, drug resistance leads to decreased efficacy and increased toxicity, resulting in unnecessary side effects [15]. Therefore, the treatment of tumors with VEGFR inhibitors alone was limited.

Mesenchymal epithelial transfer factor tyrosine kinase (c-Met) is a crucial member of the receptor tyrosine kinases (RTKs) family [16,17,18]. In normal cells, c-Met is activated by extracellular binding to its natural ligand, hepatocyte growth factor/scatter factor (HGF/SF) [19]. Many human cancers involve abnormal expression of HGF/SF or c-Met or activation of c-Met kinase mutations. The aberrant expression of c-Met/HGF signaling arises from c-Met mutations, c-Met/HGF overexpression or c-Met genomic amplification, which can promote the proliferation, migration, invasion and genesis of tumors [18,20,21]. Blocking the abnormal activation of c-Met activity is a promising method for the treatment of cancer caused by c-Met activity. Therefore, the HGF/c-Met signaling pathway has become an attractive target for tumor therapy [22,23]. At present, most small molecule inhibitors that interfere with the active site of the kinase domain have been found to be competitive inhibitors of ATP, which could block the transmission of the c-Met signaling pathway by blocking the phosphorylation of tyrosine [24]. According to their structures and binding modes with the c-Met kinase domain, small molecule inhibitors can be roughly divided into type I and type II [18,25]. Studies have shown that type I c-Met inhibitors are more selective than type II c-Met inhibitors, but type II inhibitors may be more effective than type I inhibitors since most type II inhibitors are multi-kinase inhibitors, which also have strong inhibition on VEGFR and other homologous kinases [26,27]. There is an obvious structural feature of type II c-Met inhibitors of ’5-atom regulation‘ [28,29,30]. Therefore, receptor tyrosine kinase c-Met is considered to be an essential target for the discovery of small molecule anticancer inhibitors [31,32]. The biochemical pathways of various cancers can be inhibited by drug combinations or single chemical entities with different mechanisms, which can regulate multiple targets of multifactor diseases. Multiple target inhibitors can block the transduction of multiple signal pathways, which may have a satisfactory effect on tumor therapy [33,34,35,36,37]. Thus, multi-target inhibition combined with other therapies will become a favorable approach for future clinical antitumor treatment. Among them, antitumor therapy, which is based on dual-target VEGFR and c-Met tyrosine kinase inhibitors, is a promising approach for tumor treatment [36,37,38,39,40]. The synergistic collaboration of VEGFR and c-Met promotes the development of angiogenesis and the progression of various human cancers. Thus, dual-target VEGFR/c-Met tyrosine kinase inhibitors may have more extensive advantages than VEGFR or c-Met selective inhibitors alone [37,40]. A large number of multi-kinase inhibitors have been found and applied to cancer treatment or scientific research. As shown in Table 1, these compounds, such as Foretinib, Golvatinib and Dovitinib, can act on multiple targets and have high cytotoxic activity and kinase selectivity. At present, most of the compounds have entered into clinical trials to further evaluate their pharmacodynamic (PD) and pharmacokinetic (PK) properties, in order to explore their mechanism of action and evaluate their efficacy in vivo, side effects and drug resistance. These compounds exhibited slight inhibitory effects against VEGFR and c-Met kinases, so their active skeletons are worthy of further study and may have a positive effect on the development of small anticancer inhibitors of dual-target VEGFR/c-Met kinase. Studies show that VEGFR and c-Met targeting pathways exhibit synergistic effects in inducing angiogenesis and the development of various human cancers (Figure 1). It is crucial to inhibit multiple signaling pathways, including VEGF and c-Met, to improve antitumor efficacy and overcome drug resistance [41,42]. Therefore, dual-target c-Met/VEGFR inhibitors are considered to be a promising tumor treatment method, which may be superior to c-Met selective or VEGFR selective single target inhibitors alone [43,44]. The purpose of this review is to summarize the chemical structures and biological characteristics of VEGFR/c-Met dual-target inhibitors reported in the past five years, which will help researchers to design new dual-target VEGFR/c-Met dual small molecule inhibitors.

## 2. Chemical Design Strategies of Dual-Target VEGFR/c-Met Kinase Anticancer Inhibitors

Dual-target VEGFR/c-Met kinase inhibitors are designed by combining the pharmacophores or fragments of the leading compounds acting on VEGFR and c-Met kinase targets into a molecule and modifying their skeletons. In this way, the resulting compounds have active pharmacophores or fragments for targeting VEGFR and c-Met kinases, and may have high inhibitory activity towards them [44,55]. As shown in Figure 2, the pharmacophores of VEGFR inhibitors and the pharmacophores of c-Met inhibitors are combined to form new dual-target VEGFR/c-Met inhibitors through the principles of skeleton and bioelectronic exclusion principle. They contain the main structural features of two kinase drugs, which bind to the protein sequence of VEGFR and c-Met kinases or the binding site of ligands respectively, thus blocking the signaling and bypassing pathways of VEGFR and c-Met [41,56,57]. Therefore, the development of novel dual-target VEGFR/c-Met kinase inhibitors is of great significance for future cancer treatment.

Most of the active pharmacophores of VEGFR are heterocycles containing nitrogen and sulfur atoms, such as quinolines, pyridines, thiophenes and pyridines, which are great structural fragments for hydrogen bonding with the amino acid residues of the kinase [43,58]. The active fragment of c-Met meets the ‘5-atom regulation’ of type II c-Met, which can be a flexible chain or rigid ring structure containing one or more hydrogen bond donors or receptors, so as to facilitate the formation of a hydrogen bond with the amino acid residues of the kinase [59,60]. Most of the dual-target VEGFR/c-Met inhibitors have the active pharmacophore of selective VEGFR inhibitors and the ‘5-atom’ active fragment of c-Met inhibitors so that the dual-target VEGFR/c-Met inhibitors can penetrate the ATP binding vesicles of VEGFR and c-Met receptors (PDB code: 3U6J, 3LQ8), and occupy the ATP pocket of VEGFR and c-Met binding through hydrogen bonding [57,60]. The skeleton modification of the obtained double target kinase inhibitors may be an effective way to obtain small molecular compounds, reduce toxicity and improve metabolic stability and water solubility. According to the structure of the parent cores, 11 derivatives are summarized, such as pyridines, quinolones, pyrrolopyridines, benzimidazoles and thienopyrimidines.

## 3. Dual VEGFR and c-Met Small Molecule Inhibitors

### 3.1. Pyridine Derivatives

Pyridine skeletons are widely used in VEGFR inhibitors and c-Met inhibitors. In particular, the pyridine nucleus in receptor tyrosine kinase inhibitors can extend into the ATP binding pocket of VEGFR-2 and c-Met receptors, and the nitrogen atom of the pyridine ring can interact with the amino acid residues in the hinge region. The pyridine skeleton plays a key role in the activity of small molecule antitumor inhibitors.

In 2016, Hao et al. introduced a series of aminopyrimidine derivatives and assayed their enzymatic activities against VEGFR-2 and c-Met kinases. Most target compounds have inhibition potency both on VEGFR-2 and c-Met with IC_50_ values in the nanomolar range, especially compound **1** (Figure 3). The substitution of compound **1** (VEGFR-2: IC_50_ = 0.170 ± 0.055 µM, c-Met: IC_50_ = 0.210 ± 0.030 µM) with a dimethylamino group was better than that by pyrrolidine and morpholine. As shown in Figure 4, it was found that the pyrimidine of 4-aminopyrimidine-5-formoxime could bind to the pocket of ATP in VEGFR-2 and c-Met receptor proteins. CYS-919 of the VEGFR-2 hinge region and MET-1160 of the c-Met hinge region can form hydrogen bonds with the nitrogen atoms of pyrimidine [36]. The results indicated that compound **1** was a dual inhibitor of VEGFR-2 and c-Met that holds promising potential.

A series of pyrimidine derivatives have been reported and half of the target compounds have been evaluated to exhibit moderate to potent c-Met inhibitory activities. Among these, it was noteworthy that compound **2** (Figure 3) showed most potent c-Met (IC_50_ = 17 nM) inhibitory potency. In particular, compound **2** significantly decreased the selectivity for VEGFR-2 (IC_50_ = 55 nM) kinase, which may provide an opportunity to investigate the further selectivity and reduce VEGFR-2 related side effects [19]. The results suggested that compound **2** could be a potentially interesting lead compound for the development of anticancer agents. Compared with compound **1**, the substitution of halogen chloride in the side chain of the pyridine ring has more significant kinase inhibitory activity than the long chain of compound **2**.

In our previous studies, Wang et al. screened out four series of Sorafenib derivatives bearing pyrazole scaffolds and evaluated them for cytotoxicity, and some compounds were further evaluated for kinase activities. In particular, the most promising compound, compound **3** (Figure 3), exhibited the best activity against A549, HepG2 and MCF-7 cell lines. The IC_50_ value of compound **3** on VEGFR-2 kinase was 0.56 µM. Compound **3** showed moderate to no activity against CRAF, c-Met, EGFR and FLT3 kinases. Through molecular docking simulation studies, it was found that compound **3** formed hydrogen bonds with VEGFR-2 amino acid residues CYS-919, ASP-1046 and GLU-885 [61]. Similarly, new hydrogen bonds also played a central role in improving the inhibition of pyrazole derivatives on c-Met kinase. As a potential inhibitor of VEGFR-2/c-Met kinases, further study is needed to determine its mechanism.

A series of pyridine derivatives with pyrazolone skeletons were evaluated. The cell proliferation assay in vitro showed that most of the target compounds had inhibition potency towards both VEGFR-2 and c-Met. Compound **4** (Figure 3) showed the greatest inhibitory activities against BaF3-TPR-Met and HUVEC cancer cell lines with IC_50_ values of 0.30 µM and 0.13 µM, respectively. A triazole ring formed by the cyclization of pyridylamide can improve the anti-proliferation activity of the compound, and the tertiary amine substituent shows better anti-proliferation activity, indicating that the tertiary amine has good tolerance in maintaining strong inhibitory activity. This compound also exhibited the most potent inhibitory activity against VEGFR-2 and c-Met with IC_50_ values of 0.19 µM and 0.11 µM, respectively. Molecular docking of compound **4** (Figure 5) into the ATP binding sites of c-Met and VEGFR-2 was performed and the result suggested that compound **4** could bind well with the active site of c-Met (MET-1160 and ASP-1222) and VEGFR-2 (CYS-919 and ASP-1046) [5]. These results indicated that compound **4** has become a promising candidate for further development of more potent dual-target c-Met/VEGFR-2 kinase inhibitors.

Pyridine derivatives bearing 1,6-naphthyridine scaffolds were scrutinized by Wang et al. Based on the scaffold-hopping strategy of heteroatom migration, the optimal compound **5** (MET: IC_50_ = 9.8 nM, VEGFR-2: IC_50_ = 8.8 nM) was designed. Further molecular docking showed that the cyclopropane carbonyl from compound **5** formed a good hydrophobic interaction with c-Met kinase amino acid residues TRY-1159 and LSY-1611. For VEGFR-2 kinase, the smaller steric cyclopropane carbonyl of compound **5** can be well tolerated and has a favorable pharmacokinetic profile (F = 63%, CL = 0.12 L/h/kg, and AUC_0–∞_ = 42.2 h µg/mL) [62]. The results showed that a 1,6-naphthidine scaffold could be used as a new scaffold for kinase inhibitors.

In study by Zeidan, two series of picolinamide derivatives bearing (thio)urea and dithiocarbamate were reported. All compounds were evaluated for their cytotoxic activity against A549 cancer cells. Compound **6** (Figure 3) showed enhanced potency towards VEGFR-2 (IC_50_ = 0.027 µM), c-Met (IC_50_ = 8.300 µM), EGFR (IC_50_ = 0.084 µM), HER-2 (IC_50_ = 0.104 µM), and MER (IC_50_ = 3.950 µM) kinases. Cell cycle analysis of A549 cells treated with compound **6** showed cell cycle arrest at the G2/M phase and pro-apoptotic activity [63]. Therefore, compound **6** deserves further study.

With the development of related research, new antiangiogenic drugs have been found. Biological results showed that compound **7** (Figure 3) showed significant inhibition of VEGFR-2 (IC_50_ = 2.35 nM), Tie-2 and EphB4 kinases. Compound **7** showed good selectivity for VEGFR-1 (IC_50_ = 25.00 nM), VEGFR-3 (IC_50_ = 45.00 nM) and c-Met (IC_50_ = 57.80 nM) kinases (Table 2) and was found to be a multi-target inhibitor. Biological evaluation and molecular docking showed that *N*-(pyridin-2-yl)acrylamide was beneficial to the potency of these triple inhibitors [64]. Compound **7** is a multi-kinase inhibitor with good kinase selectivity, and its role in the research of anticancer drugs is of great significance.

### 3.2. Quinoline Derivatives

Liu and co-workers used Foretinib as the lead compound and introduced 1,2,4-triazolone into the ‘5-atom’ part to synthesize a series of new 4-phenoxyquinoline derivatives. All compounds showed moderate to excellent cytotoxic activity against different cancer cells. Compound **8** (Figure 6) showed significant antitumor activity against all tested cell lines (HT-29, H460, A549, and MKN-45), especially against the HT-29 cell line (IC_50_ = 0.08 µM). In addition, compound **8** exhibited a high inhibitory effect on c-Met kinase (IC_50_ = 1.57 nM), and also showed a strong inhibitory effect on c-Kit (IC_50_ = 3.38 nM) and FLT3 (IC_50_ = 8.19 nM). In addition, compound **8** showed moderate selectivity for PDGFRα (IC_50_ = 95.23 nM), Ron (IC_50_ = 140.72 nM), and VEGFR-2 (IC_50_ = 480.47 nM) kinases (Table 3) [65]. The results indicated that compound **8** provided a new scaffold for further selectivity.

In another effort, Liu and co-workers also reported a series of novel 4-phenoxyquinoline derivatives containing 3-oxo-3,4-dihydroquinoxaline moieties. The most promising compound, compound **9** (Figure 6), has significant cytotoxicity against HT-29, H460, A54 9, MKN-45 and U87MG cell lines, with IC_50_ values of 0.06 µM, 0.05µM, 0.18µM, 0.023µM, and 0.66µM, respectively. Preliminary research also showed that compound **9** also showed a high inhibitory effect on c-Met (IC_50_ = 0.90 nM), c-Kit (IC_50_ = 2.45 nM) and PDGFRα (IC_50_ = 19.13 nM). In addition, compound **9** has moderate to excellent selectivity for Ron, VEGFR-2, FLT3, EGFR, and ALK kinases (Table 3) [22]. So, compound **9** remains a multi-target inhibitor of tyrosine kinase. Compared with compound **8**, the selectivity of VEGFR-2 kinase was improved when the ‘5-atom’ moiety of compound **9** was 3-oxo-3,4-dihydroquinoline.

During the discovery of potent VEGFR/c-Met kinase inhibitors, Nan et al. reported a series of 4-phenoxyquinoline derivatives containing sulfonylurea and evaluated them for their c-Met kinase inhibition and cytotoxicity against four cell lines. Pharmacological data indicated that most compounds showed moderate to significant potency, with the most promising compound, compound **10** (Figure 6, c-Met: IC_50_ = 1.98 nM), showing better selectivity over other tyrosine kinases (Table 3). Compound **10** had significant cytotoxicity to HT460, MKN-45, HT-29 and MDA-MB-231 cell lines, with IC_50_ values of 0.055 μM, 0.064 μM, 0.16 μM and 0.49 μM, respectively. Docking studies show that the introduction of sulfonylurea into the structure maintains the strong cytotoxicity of the compounds [18].

Compound **11** (Figure 6) had good pharmacokinetic properties in vivo, had no obvious AO-mediated metabolite in a xenotransplantation model of H1993 and SNU-5 with high expression of c-Met, and has good antitumor effect. The selectivity of compound **11** was studied with human tyrosine kinases. The results showed that compound **11** had high selectivity to c-Met, and the IC_50_ values of the other tested tyrosine kinases were higher than 700 nM [66].

### 3.3. Pyrrolopyridine Derivatives

In our previous studies, Wang et al. screened two series of aromatic hydrazone derivatives bearing a 1H-pyrrolo[2,3-*b*]pyridine moiety and evaluated them for cytotoxicity; some compounds were further evaluated for kinase activities. Most of the compounds showed moderate to extremely good cytotoxic activity towards four cancer cell lines (A549, HepG2, MCF-7 and PC-3 cells). Among them, the inhibitory activities of compounds **12** (Figure 7) and **13** (Figure 7) were better than those of positive control (Foretinib). Compounds **12** and **13** showed moderate inhibition on c-Met kinase, and the activity of compound **12** (IC_50_ = 0.506 μM) was slightly better than that of compound **13** (IC_50_ = 0.907 μM), similar to the cytotoxicity results. The activity data showed that compounds **12** and **13** had selective inhibition on c-Met (IC_50_ = 0.5 μM, and 0.9 μM) compared with FLT3 (IC_50_ = 2.6 μM, and 3.8 μM), VEGFR-2 (IC_50_ = 8.9 μM, and 14.4 μM) and EGFR (IC_50_ > 100 μM, and IC_50_ = 45.0 μM) kinases. Therefore, compounds **12** and **13** can be considered as multi-target small molecule inhibitors [67].

Wang et al. used Foretinib as the lead compound and a pyrrolo[2,3-*b*]pyridine moiety as the core, and then introduced 4-oxopyridazinone into the ‘5-atom’ moiety to limit the conformation. Finally, the optimal compound **14** was found. It had significant cytotoxicity against A549 (IC_50_ = 2.190 μM), HepG2 (IC_50_ = 1.320 μM), MCF-7 (IC_50_ = 6.270 μM) and PC-3 (IC_50_ = 4.630 μM) cell lines in vitro (Table 4), and it showed good kinase selectivity for FLT3 (IC_50_ = 0.770 μM), VEGFR-2 (IC_50_ = 2.400 μM), c-Kit (IC_50_ = 2.200 μM) and EGFR (IC_50_ > 10.000 μM) kinases (Table 5). Therefore, compound **14** can also be considered as a multi-target small molecule inhibitor [68].

### 3.4. Benzimidazole Derivatives

Benzimidazole is a privileged lead nucleus which exists in many anticancer drugs. Moreover, some reports showed that the anticancer activities of some benzimidazole derivatives were mediated by targeting VEGFR-2. Some benzimidazole derivatives, such as benzimidazolyl amino quinoline derivatives, exhibited dual inhibition of VEGFR-2 and c-Met kinases.

Three series of benzimidazole derivatives which were hybridized with piperazine, oxadiazole and triazolo-thiadiazole, respectively, were synthesized. Compound **15** (Figure 8) had good cytotoxic activity on HCT-116 (IC_50_ = 2.19 ± 0.09 µM) and A549 (IC_50_ = 10.97 ± 0.09 µM) cells (Table 5). Compound **15** exhibited the best activity against VEGFR-2 and c-Met kinases, with an inhibition percent of 29.22% for VEGFR-2 and 71.66% for c-Met, which were close to those of the reference inhibitor AG-1478 (VEGFR-2: 73.73%, c-Met: 33.12%). The selectivity for c-Met kinase was stronger than that for VEGFR-2 kinase. Molecular simulation studies showed that the thiadiazole ring of compound **15** interacted with the amino acid residue CYS-919 of VEGFR-2 kinase to form a hydrogen bond (Figure 9). The sulfur atom of thiadiazole in compound **15** formed a hydrogen bond interaction with the amino acid residue ASP-1222 of c-Met kinase. The docking results indicated that the addition of triazolothiazole fragments enhanced the binding of compound **15** to the receptor, which may enhance the binding affinity and enhance the antitumor activity of the compound [69]. Therefore, compound **15** may be a good dual-target VEGFR-2/c-Met small molecule inhibitor.

Ibrahim et al. also reported in the same year that the docking results of compound **16** (Figure 8) were related to its moderate enzyme inhibitory activity on VEGFR-2 and its good inhibitory activity on c-Met (35.88% and 88.48%, respectively), as shown in Table 5. The inhibitory rates of compound **17** (Figure 8) on NCI-H522 and SK-MEL-2 were 48.70% and 42.62%, respectively, and the inhibitory activities on VEGFR-2 and c-Met kinases were 35.88% and 82.48% (Table 6), respectively. The binding mode of compound **16** with VEGFR-2 and c-Met was related to the data from the kinase assay, which indicated that compound **17** is a good dual-target VEGFR-2/c-Met small molecule inhibitor [70].

### 3.5. Thienopyrimidine Derivatives

The thienopyrimidine skeleton had been widely used in FGFR1 inhibitors, B-Raf inhibitors, FLT3 inhibitors and EGFR kinase inhibitors. In 2017, Li et al. identified a series of thienopyrimidine derivatives as novel dual-target VEGFR-2/c-Met kinase inhibitors. The inhibition rates of compound **18** (Figure 10) on HUVEC and BaF3-TPR-Met cell lines were 79.52% and 95.97%, respectively. Cell analysis showed that the introduction of amides in the solvent area would improve the compounds’ cell cytostatic activity, and that long chain polar substituents made the compounds more tolerant. Therefore, the introduction of polar flexible chains in the solvent area are conducive to improving the inhibition rate of the compounds on tumor cells. Further evaluation of the kinase activity of the compounds towards VEGFR-2 and c-Met was carried out using ELISA. The result showed that most of the target compounds had inhibition potency towards VEGFR-2 and c-Met with IC_50_ values in the nanomolar range. Among the test compounds, compound **18** (which was found to be the most potent compound) could inhibit the kinase activities of VEGFR-2 and c-Met with IC_50_ values of 0.048 ± 0.006 µM and 0.025 ± 0.003 µM, respectively. This study provided us with clear SARs, which would help design more effective VEGFR-2/c-Met inhibitors [71].

### 3.6. Pyrrolotriazine Derivatives

Pyrrole [1,2-*f*] [1,2,4] triazine has been widely used in kinase domains. In 2018, Wei et al. designed and synthesized a series of novel pyrrolotriazine double c-Met/VEGFR-2 kinase derivatives. Compound **18** showed strong antiproliferative activity in BaF3-TPR-Met cells (IC_50_ = 0.710 ± 0.160 nM) and HUVEC cells (IC_50_ = 3.740 ± 0.311 nM), and inhibited the phosphorylation of VEGFR-2 and c-Met and the corresponding downstream signaling pathways (MAPK and PI3K). Compound **19** (Figure 10) has great selectivity to c-Met and VEGFR-2, and potent inhibitory activity against them (IC_50_ = 2.300 ± 0.100 nM and 5.000 ± 0.500 nM). Western blotting was used to study the inhibitory effect of compound **19** on VEGFR-2 activation and the MAPK/PI3K signaling pathway in HUVEC cells. Flow cytometry analysis showed that different concentrations of compound **19** (1 nM and 10 nM) significantly inhibited the proliferation of BaF3-TPR-Met cells, and the percentage of cells in the G0/G1 phase increased in a concentration dependent manner, which may be achieved by periodically blocking and inducing apoptosis in the G0/G1 phase. Compound **19** has good pharmacodynamics and physicochemical properties. In addition, docking experiments showed that compound **19** occupied the ATP pocket combined with VEGFR-2 (CYS-919 and ASP-1046) and c-Met (MET-1160, ASP-1222, LYS-1161 and LYS-1110) kinases through hydrogen bonding [56]. The evaluation experiments indicated that compound **19** was a potential dual-target c-Met/VEGFR-2 kinase inhibitor for cancer therapy deserving further study.

### 3.7. Quinazoline Derivatives

Quinazoline drugs can inhibit a variety of tyrosine kinases, including c-kit, c-Met, VEGFR, EGFR, PDGFR, FGFR, and so on. Some quinazoline derivatives are highly effective antitumor drugs, such as Cabozantinib and Fortinib. Therefore, quinazoline derivatives are promising antitumor drugs, especially for targeting c-Met/VEGFR-2 kinases.

Recently, [1,4]dioxa[2,3-*f*]quinazoline derivatives were scrutinized by Deng et al. in order to explore their potential as VEGFR-2/c-Met kinase inhibitors. Compounds **20** and **21** (Figure 11) exhibited potent inhibitory activity against VEGFR-2 (IC_50_ = 4.8 nM and 3.5 nM) and c-Met (IC_50_ = 5.8 nM and 7.3 nM) kinases. Importantly, compound **20** showed excellent antiproliferative activities on MHCC97H and HUVEC cells with IC_50_ values of 15.7 nM and 0.8 nM, respectively. As shown in Figure 12A, compounds **20** and **21** formed hydrogen bonds with the ASP-1064 residue on the binding site of VEGFR-2 kinase, and the second nitrogen atom of quinazoline formed hydrogen bonds with the CYS-919 residue. Figure 12B showed that compounds **20** and **21** were linked to the binding site of c-Met kinase, and quinazoline formed hydrogen bond interactions with MET-1160 [72]. All results indicated that compounds **20** and **21** were dual-target VEGFR-2/c-Met kinase inhibitors that held promising potential in cancer therapy.

### 3.8. Diazepine Derivatives

In recent years, a series of 6,11-dihydro-5*H*-benzo[*e*]pyrimido-[5,4-*b*][1,4]diazepine derivatives have been evaluated for their inhibition of c-Met kinase. In addition to the kinase activity of c-Met (IC_50_ = 24.4 nM), compound **22** (Figure 11) also has strong inhibition on other kinases such as VEGFR-2 (IC_50_ = 62.5 nM), EGFR (IC_50_ = 267.6 nM), RET (IC_50_ = 162.8 nM), c-Kit (IC_50_ = 258.7 nM) and FLT3 (IC_50_ = 851.8 nM). These results indicated that compound **22** was a promising multi-target inhibitor of tyrosine kinase. It also showed good pharmacokinetic characteristics in rats, had acceptable safety in preclinical research, and had significant antitumor activity in a Caki-1 tumor xenotransplantation model [73]. Compound **22** provided a new scaffold for further optimization.

### 3.9. Pyrazolopyrimidine Derivatives

A series of 1*H*-pyrazolo[3,4-*d*]pyrimidine derivatives were reported and evaluated for their BRAF_V600E_ and VEGFR-2 inhibitory activity as well as their anti-proliferative activity. Among them, compound **23** (Figure 13) had high inhibition towards BRAF_V600E_ (IC_50_ = 0.171 μM) and VEGFR-2 (IC_50_ = 0.779 μM), and a good anti-proliferation effect on A375, HT-29 and HUVEC cell lines. Moreover, compound **23** blocked A375 and HUVEC cells mainly in the G0/G1 phase. Compound **23** showed moderate inhibition on VEGFR-1 and VEGFR-3 kinases, but no significant inhibition on c-Met kinase [74]. The results showed that compound **23** had good selectivity and was a multi-kinase inhibitor.

### 3.10. Naphthyridinone Derivatives

Zhuo et al. developed a series of 2,7-naphthyridone-based derivatives of BMS-777607 as new MET kinase inhibitors. Among them, compound **24** (Figure 13) had significant c-Met (IC_50_ = 12.6 nM) and VEGFR-2 (IC_50_ = 17.3 nM) kinase inhibitory activities [2]. It is shown that 2,7-naphthyridone can be used as a novel dual-target antitumor drug scaffold.

### 3.11. Triazine Derivatives

Tetrahydrobenzo[*b*]thiophene derivatives have been found in the study of multi-target antitumor drugs. The most promising compound complexes **25** (Figure 13) and **26** (Figure 13) had significant c-Met kinase (IC_50_ = 0.42 nM and 0.49 nM, respectively) inhibitory activity. They also had inhibitory activity towards c-Kit (IC_50_ = 0.62 nM and 0.16 nM, respectively), FLT3 (IC_50_ = 0.49 nM and 0.24 nM, respectively), VEGFR-2 (IC_50_ = 0.26 nM and 0. 24 nM, respectively), EGFR (IC_50_ = 0.38 nM and 0.49 nM, respectively) and PDGFα (IC_50_ = 0.41 nM and 0.22 nM, respectively) kinases [75]. Therefore, compounds **25** and **26** are potential multi-target antitumor inhibitors.

## 4. Structure–Activity Relationship

The structural characteristics of most dual-target VEGFR/c-Met inhibitors are shown in Figure 14, and they can be divided into four parts: A, B, C and D. In part A, different nitrogen-containing aromatic heterocycles can be introduced, and substituents or heterocycles containing other heteroatoms such as nitrogen and sulfur atoms can be added to form substituted pyridine, pyrrolidine, quinazoline and other structures. When the side chain of the A-ring of pyridine derivatives is replaced by an amide chain, they show better activity. The side chains of quinoline and quinazoline derivatives are mostly substituted by alkyl oxygen groups, which can change the water solubility of the compounds and have little effect on the activity. If the side chains of the pyrrolidine derivative ring are replaced by a lipophilic group, its activity may be improved. Part A is a major structural fragment which formed a hydrogen bond with the amino acid residues of VEGFR (CYS-919) and c-Met (MET-1160) kinases (Figure 15). Part B is mainly composed of a pyridine ring and a benzene ring with various substitutions or not. Part C usually meets the ‘5-atom regulation’. In part C, a flexible chain or rigid ring structure such as pyrazolone, naphthyridine or triazolo-thiadiazole can be introduced. Urea structures contain hydrogen bond donor or acceptor atoms for forming hydrogen bonds with residues of VEGFR (LYS-868, ASP-1046, etc.) and c-Met (ASP1220, LYS1110, LEU1245, etc.) kinases (Figure 15). A new five atom connection bridge is also formed so that part D can be fully embedded in the hydrophobic pocket (Figure 15). When part D is mainly composed of benzyl or fluorophenyl, the activity of the compound is better. Among other compounds which do not conform to the five atom rule, tetrahydrobenzothiophene derivatives with a triazine structure have better activity.

## 5. Conclusion and Perspectives

So far, dozens of small molecule kinase inhibitors have been approved for the treatment of different types of human cancer, but the emergence of drug resistance and toxicity caused by single-agent treatment or combinations of drugs are the main challenges faced by cancer patients. Studies have shown that multi-target inhibitors can combine with different targets simultaneously, regulate multiple signaling pathways and control the development of diseases to increase the therapeutic effect. Therefore, the development of dual-target inhibitors is of great significance for cancer treatment in the future.

In addition, this review not only summarizes the dual-target VEGFR/c-Met small molecule antitumor inhibitors with significant cytotoxicity and kinase inhibitory activity published in recent years, but also shows some active skeletons and fragments. These active skeletons and fragments may play an role in enhancing the inhibitory activity of antitumor derivatives, reducing toxicity or improving the stability and water solubility of compounds. It can be clearly seen from the above that the research and exploration of dual-target kinase compounds have a positive effect on the development of drugs for the treatment of various cancers. Other types of dual-target drugs are also emerging in current cancer treatments. Multi-target inhibition combined with other therapies will become a considerable future clinical antitumor treatment, and more small molecule antitumor inhibitors with higher potency and selectivity will appear.

## Figures and Tables

**Figure 1 molecules-25-02666-f001:**
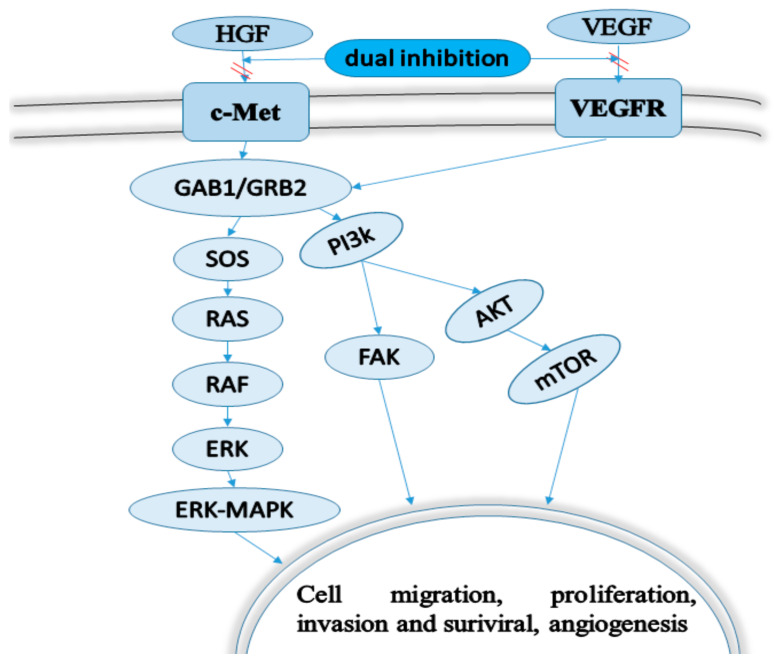
The crosstalk between VEGFR and c-Met (GAB1: GRB2-associated-binding protein 1; GRB2: Growth factor receptor-bound protein 2; SOS: Son of sevenless; RAS: Rat sarcoma viral oncogene homolog; RAF: Root abundant factor; FAK:Focal adhesion kinase; PI3K/AKT: Phosphoinositide 3-kinase/protein kinase B; mTOR:Mammalian target of rapamycin/mechanistic target of rapamycin.).

**Figure 2 molecules-25-02666-f002:**
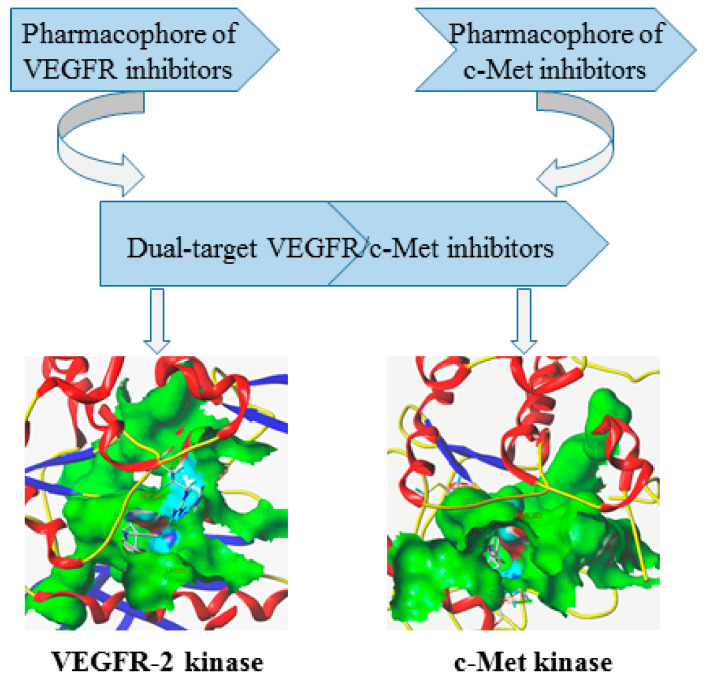
Design of dual-target VEGFR/c-Met kinase inhibitors (PDB code: 3U6J, 3LQ8).

**Figure 3 molecules-25-02666-f003:**
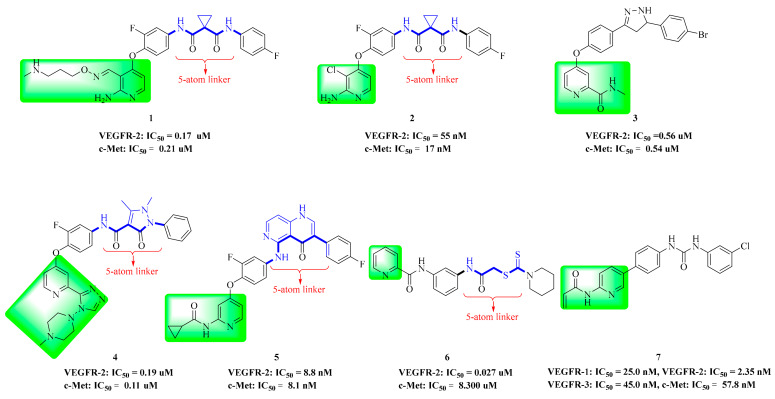
Pyridine derivatives of dual VEGFR/c-Met inhibitors.

**Figure 4 molecules-25-02666-f004:**
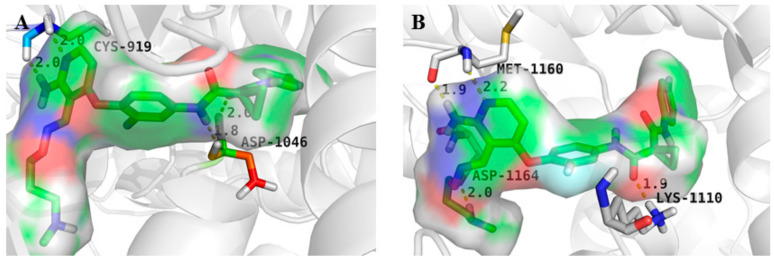
(**A**) The model of compound **1** binding to VEGFR-2 (PDB code: 3U6J); (**B**) The model of compound **1** binding to c-Met (PDB code: 3LQ8).

**Figure 5 molecules-25-02666-f005:**
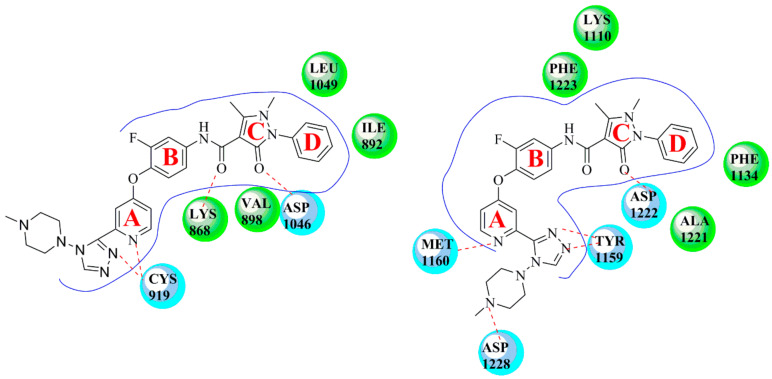
Docking of compound **4** with VEGFR-2 (PDB code: 3U6J) and c-Met (PDB code: 3LQ8) (Most dual-target VEGFR/c-Met inhibitors can be divided into four parts: A, B, C and D. Part A: The main structural skeleton, mainly nitrogen-containing heterocycles. Part B: Substituted benzene ring; Part C: Meet the ‘5-atom regulations’. Part D; Benzene ring).

**Figure 6 molecules-25-02666-f006:**
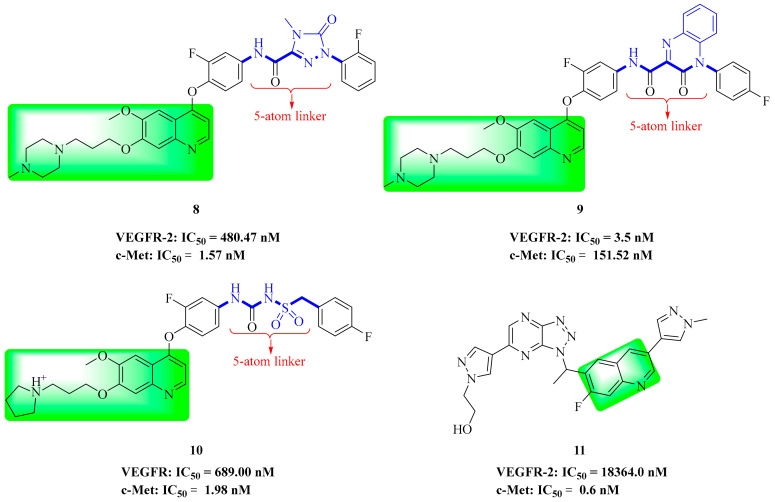
Quinoline derivatives of dual VEGFR/c-Met inhibitors.

**Figure 7 molecules-25-02666-f007:**
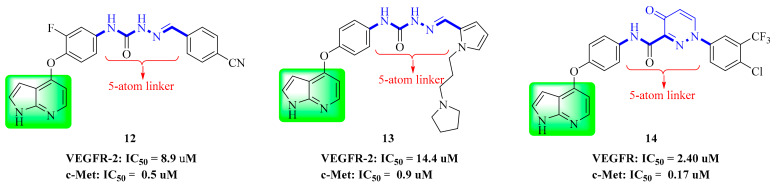
Pyrrolopyridine derivatives of dual VEGFR/c-Met inhibitors.

**Figure 8 molecules-25-02666-f008:**
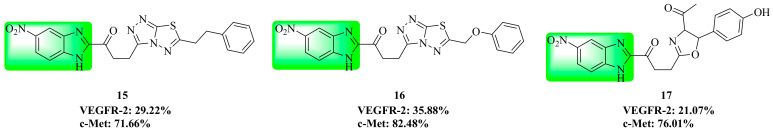
Benzimidazole derivatives of dual VEGFR/c-Met inhibitors.

**Figure 9 molecules-25-02666-f009:**
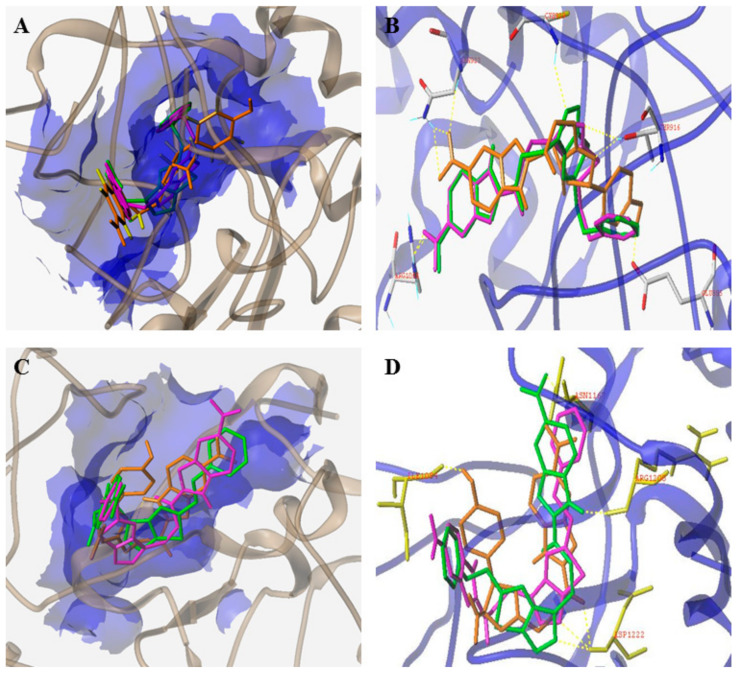
(**A**,**B**) The model of compounds **15** (magenta), **16** (green) and **17** (orange) bound to VEGFR-2 (PDB code: 2QU5); (**C**,**D**). The model of compounds **15**, **16** and **17** bound to c-Met (PDB code: 3CD8).

**Figure 10 molecules-25-02666-f010:**
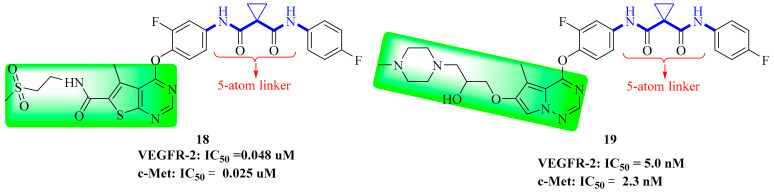
Thienopyrimidine and pyrrolotriazine derivatives of dual VEGFR/c-Met inhibitors.

**Figure 11 molecules-25-02666-f011:**
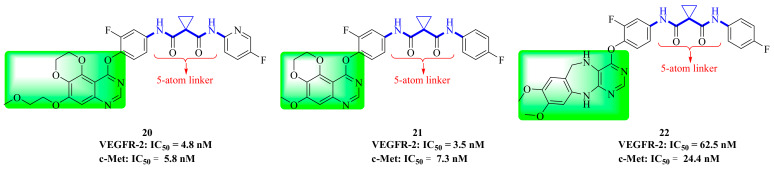
Quinazoline and diazepine derivatives of dual-target VEGFR/c-Met inhibitors.

**Figure 12 molecules-25-02666-f012:**
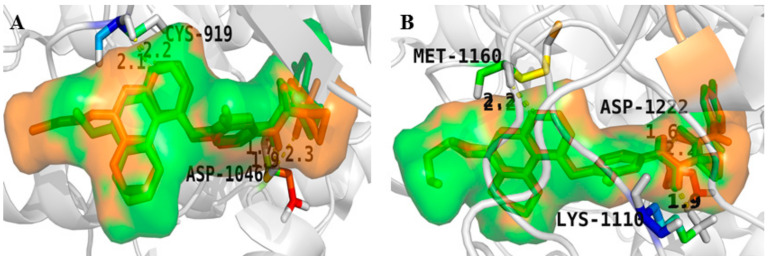
(**A**) The model of compounds **20** (green) and **21** (orange) bound to VEGFR-2 (PDB code: 3U6J); (**B**) The model of compounds **20** and **21** bound to c-Met (PDB code: 3LQ8).

**Figure 13 molecules-25-02666-f013:**
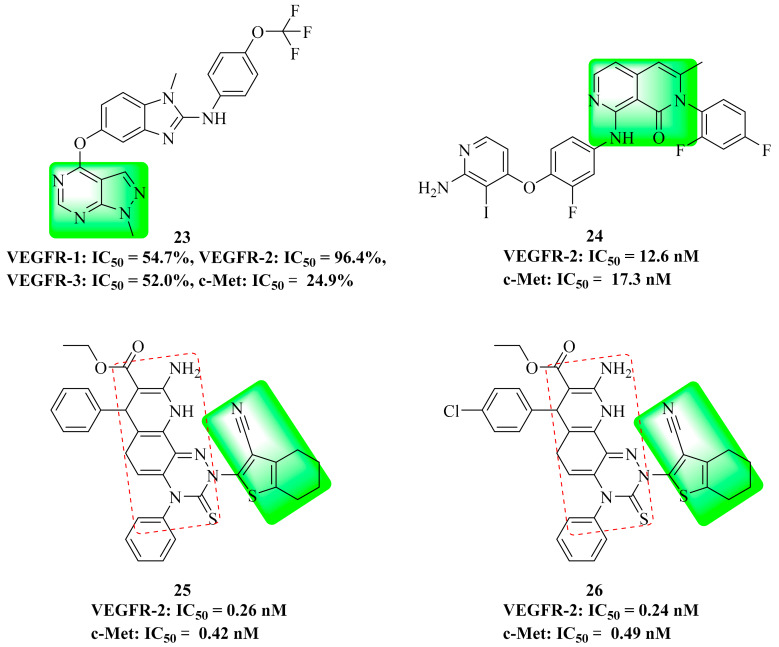
Structures of dual VEGFR/c-Met inhibitors.

**Figure 14 molecules-25-02666-f014:**
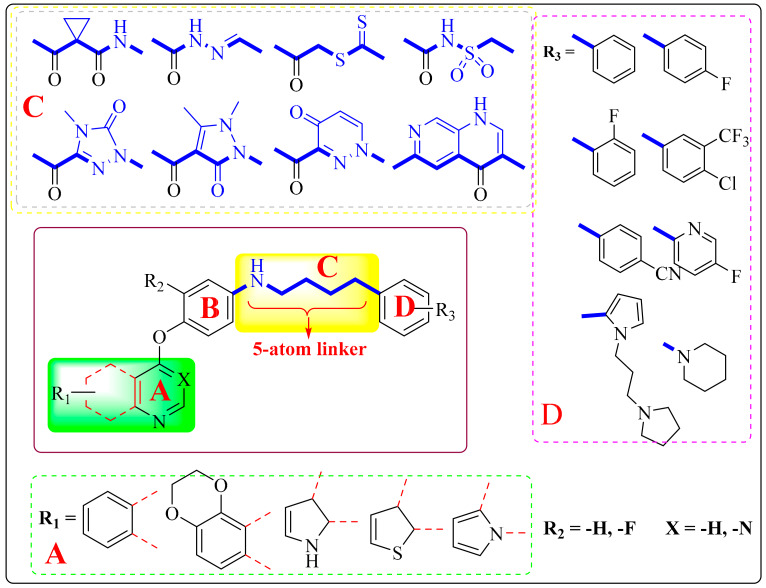
General structural formula of dual-target VEGFR/c-Met kinase inhibitors (Most dual-target VEGFR/c-Met inhibitors can be divided into four parts: A, B, C and D. Part A: The main structural skeleton, mainly nitrogen-containing heterocycles. Part B: Benzene ring or substituted benzene ring; Part C: Usually meet the ‘5-atom regulations’. Part D; Mostly benzene ring, substituted benzene ring or substituted pyridine ring).

**Figure 15 molecules-25-02666-f015:**
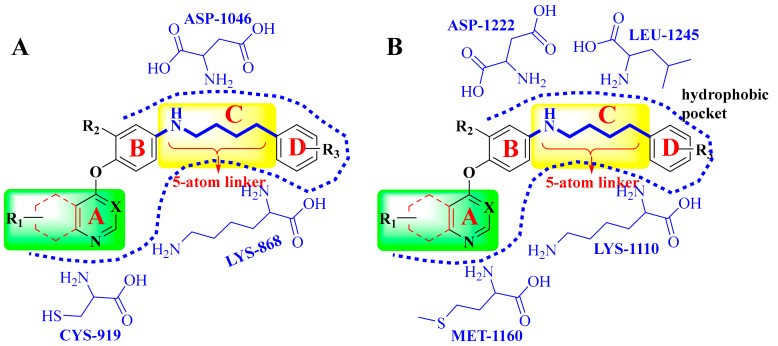
(**A**) The model of dual-target VEGFR/c-Met inhibitors bound to VEGFR. (**B**) The model of dual-target VEGFR/c-Met inhibitors bound to c-Met.

**Table 1 molecules-25-02666-t001:** Multi-target vascular endothelial growth factor receptor/mesenchymal epithelial transfer factor tyrosine kinase (VEGFR/c-Met) inhibitors in clinical trials.

Compounds	Structural Formula	Target	Indication	Phase	Signaling Pathway	Research and Development Company
Foretinib [45]	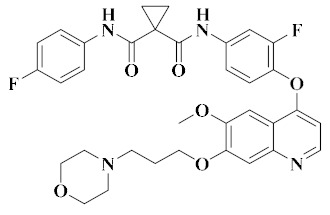	c-Met, VEGFR-2 (KDR), Tie-2, VEGFR-3/FLT4	Gastric cancer and head/neck cancer	II	Protein tyrosine kinase	Exelixis
Golvatinib [46]	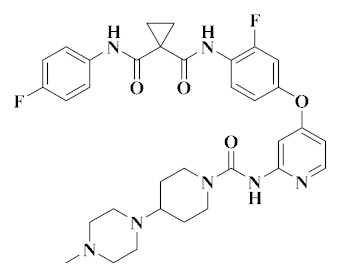	c-Met, VEGFR-2	Head and neck cancer, liver cancer	II	Angiogenesis; protein tyrosine kinase	Eisai
Dovitinib [47]	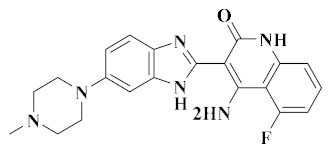	FLT3, c-Kit, FGFR-1/3, VEGFR1-4, EGFR, c-Met	Solid tumors	IV	Angiogenesis	Novartis
Tivozanib [48]	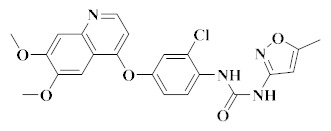	VEGFR-1, VEGFR-2, VEGFR-3, c-Met, PDGFR, c-Kit	Advanced renal cell carcinoma	III	Angiogenesis	Aevo
BMS-794833 [49]	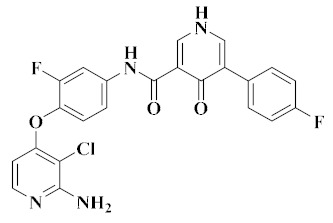	c-Met, VEGFR-2, Ron, Axl, FLT3	Gastric cancer	I	Angiogenesis; protein tyrosine kinase	Bristol Myers Squibb
BMS-777607 [50]	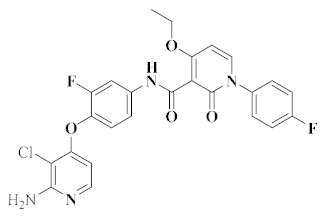	c-Met, Axl, Ron, VEGFR-2, lck	Advanced solid tumors	II	Protein tyrosine kinase	Bristol Myers Squibb
MGCD-265 [51]	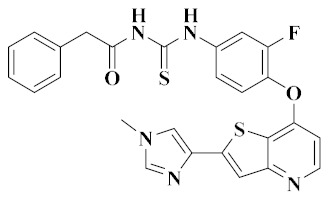	c-Met, Ron, VEGFR-1, VEGFR-2	Non-small cell lung cancer	II	Angiogenesis; protein tyrosine kinase	MethylGene
AC480 [52]	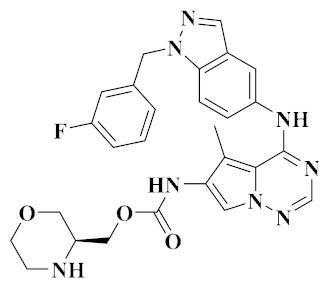	HER1, HER2, HER4, VEGFR-2, c-Kit, Lck, MET	Advanced solid tumors	I	Angiogenesis	Ambit Biosciences
CP-724714 [53]	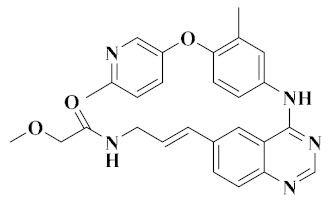	HER2/ErbB2, EGFR, VEGFR-2, c-Met	Advanced solid tumors	II	Protein tyrosine kinase	Pfizer
AMG-458 [54]	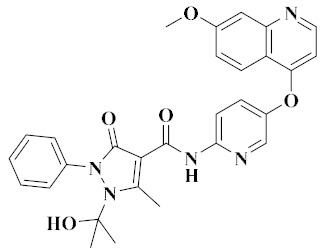	c-Met, VEGFR-2	Solid tumors	Non-medicinal	Protein tyrosine kinase	Amgen

**Table 2 molecules-25-02666-t002:** RTKs selectivity profile of compound **7.**

Compound	IC_50_ (nM)							
	VEGFR-2	c-Met	VEGFR-1	VEGFR-3	EGFR	IGF1-R	B-Raf	c-Kit
7	2.35	33.12	25.00	45.00	8.70	40.00	32.00	68.90

**Table 3 molecules-25-02666-t003:** Inhibition of tyrosine kinases by compounds **8**, **9** and **10.**

Compound	IC_50_ (nM)							
	c-Met	c-Kit	FLT3	PDGFRα	Ron	VEGFR	EGFR	ALK
8	1.57	3.38	8.19	95.23	140.47	480.47	>10000	>10000
9	0.09	2.45	268.81	19.13	82.56	151.52	980.83	2840.72
10	1.98	380	400	242	375	689	>10000	>10000

**Table 4 molecules-25-02666-t004:** Inhibition of cells by compounds **12**, **13** and **14.**

Compound	IC_50_ (μM)			
	A549	HepG2	MCF-7	PC-3
12	0.82	1.00	0.93	0.92
13	1.30	1.42	4.53	4.29
14	2.19	1.32	6.27	4.63

**Table 5 molecules-25-02666-t005:** Inhibition of kinases by compounds **12**, **13** and **14**.

Compound	IC_50_ (μM)				
	c-Met	VEGFR-2	c-Kit	FLT3	EGFR
12	0.5	8.9		2.6	>100
13	0.9	14.4		3.8	45.0
14	0.073	2.4	2.2	0.77	>10

**Table 6 molecules-25-02666-t006:** Inhibition percent of compounds **15**, **16** and **17** for VEGFR-2/c-Met kinases.

Compound	% Inhibition at 10 µM	
	VEGFR-2	c-Met
15	29.22	71.66
16	35.88	82.48
17	21.07	76.01
AG-1478	73.73	33.12

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
