# Peer review of "Research Progress of Small Molecule VEGFR/c-Met Inhibitors as Anticancer Agents (2016–Present)"

_molecules, 2020, doi:10.3390/molecules25112666_

Round 1

Reviewer 1 Report

The review by Zhang et al is well written and updated. I have a general comment and some minor observations, though.

General comment: the Section 3 describes in a comprehensive manner the potential dual-inhibitor candidates, but does not emphasize the differences, when present, in the strength of the data that support the conclusions drawn. In some cases, data are available on cell lines, in some case they are not; in some cases, pharmacodynamics data are available, in some cases they are not.  I think that the authors can better underline these aspects, which can help the reader in understanding which are the most solid data, and can dampen the “telephone list” feeling that a reader can have.

I have also some minor comments:

-There are some run off sentences that can shortened or splitted into two, or reformulate to increase readability: lines 71-73, lines 263-64, lines 268-69

-Claim in line 30 has to be supported by at least a reference.

-Line 121 repeats the purpose of the review already statedin line 92, and can be dropped.

-There is no reference that supports what authors state in the lines 135-144, and in the figure 3. A ref is reported ( n.32) but it is not Hao et al, as it should be.

For a matter of readability, please change the measure unit of IC50  of line 307 from um to nM, as in the remaining part of the manuscript.

Author Response

Reviewer #1: The review by Zhang et al is well written and updated. I have a general comment and some minor observations, though.

General comment: the Section 3 describes in a comprehensive manner the potential dual-inhibitor candidates, but does not emphasize the differences, when present, in the strength of the data that support the conclusions drawn. In some cases, data are available on cell lines, in some case they are not; in some cases, pharmacodynamics data are available, in some cases they are not.  I think that the authors can better underline these aspects, which can help the reader in understanding which are the most solid data, and can dampen the “telephone list” feeling that a reader can have.

R: Thank you for your constructive suggestions. We have modified the content of Section 3 according to your suggestions to make the content more abundant and also help readers understand which data is reliable. For example, lines 248-253 in the original manuscript has been substantially revised, and other parts have also been revised to some extent.

I have also some minor comments:

  1. There are some run off sentences that can shortened or splitted into two, or reformulate to increase readability: lines 71-73, lines 263-64, lines 268-69

R: Thank you for your significant reminding and suggestion. We have made changes in accordance with your suggestions and conducted a full text review. We tried our best to find as many grammatical and spelling mistakes as possible and then corrected them.

  1. Claim in line 30 has to be supported by at least a reference.

R: Thank you very much for reading my manuscript carefully. References have been added on line 30 according to your suggestion.

  1. Line 121 repeats the purpose of the review already statedin line 92, and can be dropped.

R: Thanks to reviewer for reminding and suggestion. We have deleted the duplicate content in the manuscript.

  1. There is no reference that supports what authors state in the lines 135-144, and in the figure 3. A ref is reported ( n.32) but it is not Hao et al, as it should be.

R: Thank you for your kindly reminding. Due to an error in the inserted references 27-28, the serial numbers of the following references do not match the references cited in the manuscript. We have carefully checked and corrected.

  1. For a matter of readability, please change the measure unit of IC50 of line 307 from um to nM, as in the remaining part of the manuscript.

R: Thanks to reviewer for reminding and suggestion. We have kept the units of IC50 values in the manuscript consistent.

Reviewer 2 Report

In their review article, Zhang et al. are summarizing chemical structures and inhibitory potencies of distinct classes of small molecule kinase inhibitors that have the potential to target both the VEGFR2 and c-MET kinases. These two receptors are prominent targets in anti-cancer therapeutic approaches and their inhibition is being investigated in preclinical and clinical studies already for several years.

Zhang et al. are presenting a summary of recent attempts concerning synthesis of small molecule multi-kinase inhibitors and are addressing their potential utility as dual MET/VEGFR2 inhibitors. Despite some interesting insights and discussions of these findings (e.g., Chapter 4), the entire manuscript is very descriptive, technical and thus not very engaging to read. Many of the figures and tables could be presented more efficiently and also the text itself could be economized in order to deliver the necessary information and without forcing the reader to laboriously search for a piece of information in a long stretch of words. Also, a more concrete rationale of dual MET/VEGFR2 targeting and relevant literature should be discussed instead of very general and not accurate statements (e.g., statement on lines 76-77 is an overstatement in this respect).

Additional comments:

  1. There are much too many grammar errors and misspellings (singular and plural of verbs, sentences without meaning, line 112 – twice pyridine, etc.) – the entire text need to be carefully edited with respect to correct use of English.
  2. Sentences and statements in the Introduction are very repetitive and general, without a clear idea of a message.
  3. Chapter 2 does not deliver enough information and it is questionable if this part of the text should be a chapter on its own.
  4. Figure 3 is referred to in the text only after Figure 4 – perhaps the order of these two figures should be switched.
  5. Overall, there are parts of the text where it is not clear to which previously published study the text refers, for example page 6, lines 165 and on.
  6. Table 2 is not very informative and it should be reconsidered if the data presented within this single-lined table could not be incorporated into the text or as into another table presenting other inhibitors as well.
  7. Although the manuscript aims at discussing possible dual MET/VEGFR2-targeting molecules, some of the subchapters (e.g., 3.2 or 3.3) do not match this selection and are referring for example to multikinase inhibitory compounds.

Visualization of Table 4 should be improved – it is rather strange to mix within a table line values on cell lines on one hand and particular kinases on the other.

Author Response

Reviewer #2: In their review article, Zhang et al. are summarizing chemical structures and inhibitory potencies of distinct classes of small molecule kinase inhibitors that have the potential to target both the VEGFR2 and c-MET kinases. These two receptors are prominent targets in anti-cancer therapeutic approaches and their inhibition is being investigated in preclinical and clinical studies already for several years.

Zhang et al. are presenting a summary of recent attempts concerning synthesis of small molecule multi-kinase inhibitors and are addressing their potential utility as dual MET/VEGFR2 inhibitors. Despite some interesting insights and discussions of these findings (e.g., Chapter 4), the entire manuscript is very descriptive, technical and thus not very engaging to read. Many of the figures and tables could be presented more efficiently and also the text itself could be economized in order to deliver the necessary information and without forcing the reader to laboriously search for a piece of information in a long stretch of words. Also, a more concrete rationale of dual MET/VEGFR2 targeting and relevant literature should be discussed instead of very general and not accurate statements (e.g., statement on lines 76-77 is an overstatement in this respect).

R: Thanks for your constructive suggestion. The entire manuscript has been revised by us, and some general and inaccurate descriptions have been corrected. And in order to facilitate readers to read, added, we also added some Figures and Tables.

Additional comments:

  1. There are much too many grammar errors and misspellings (singular and plural of verbs, sentences without meaning, line 112 – twice pyridine, etc.) – the entire text need to be carefully edited with respect to correct use of English.

R: Thank you for your kindly reminding. We tried our best to find as many grammatical and spelling mistakes as possible and then corrected them.

  1. Sentences and statements in the Introduction are very repetitive and general, without a clear idea of a message.

R: Thank you for your constructive suggestion. Some general and repetitive sentences in the manuscript have been corrected and deleted.

  1. Chapter 2 does not deliver enough information and it is questionable if this part of the text should be a chapter on its own.

R: Thank you very much for reading our manuscript carefully. Chapter 2 mainly describes the chemical design strategies of dual-target VEGFR/c-Met anti-tumor inhibitors in recent years. We believe that it is necessary to list this chapter as a separate chapter. On the one hand, it can make readers understand the general design ideas of dual-target VEGFR/c-Met inhibitors. On the other hand, it can make readers know more information about dual-target VEGFR/c-Met inhibitors. We have also made some changes to the content of this chapter.

  1. Figure 3 is referred to in the text only after Figure 4 – perhaps the order of these two figures should be switched.

R: Thank you very much for your suggestion, but we think it is not necessary. Because Figure 3 summarizes all the pyridine derivatives in the manuscript, and Figure 4 is only the molecular docking diagram of compound 1.

  1. Overall, there are parts of the text where it is not clear to which previously published study the text refers, for example page 6, lines 165 and on.

R: Thank you very much for reading our manuscript carefully. Due to an error in the inserted references 27-28, the serial numbers of the following references do not match the references cited in the manuscript. We have carefully checked and corrected.

  1. Table 2 is not very informative and it should be reconsidered if the data presented within this single-lined table could not be incorporated into the text or as into another table presenting other inhibitors as well.

R: Thanks to reviewers for reminding and suggestion. We did not delete Table 2, because it can intuitively show that compound 7 is a multi-kinase inhibitor, shows significant inhibitory activity against VEGFR/c-Met kinase and shows that compound 7 has kinase selectivity. However, we have partially modified the description of compound 7.

  1. Although the manuscript aims at discussing possible dual MET/VEGFR2-targeting molecules, some of the subchapters (e.g., 3.2 or 3.3) do not match this selection and are referring for example to multikinase inhibitory compounds.

R: Thanks for your comments on our work. Although some compounds in the manuscript are multi-kinase inhibitors, which seem to be inconsistent with the title of this article, most of the dual-target VEGFR/c-Met kinase inhibitors have certain inhibitory activities against multiple kinases. These compounds showed significant inhibitory activities against VEGFR and c-Met kinases, and to some extent also showed kinase selectivity.

  1. Visualization of Table 4 should be improved – it is rather strange to mix within a table line values on cell lines on one hand and particular kinases on the other.

R: Thanks for your constructive suggestion. We have made changes to Table 4 according to your suggestions, and the contents of the original Table 4 are now presented in Table 4 and Table 5.

Reviewer 3 Report

In my opinion the paper merits publication in Molecules.

In some sentences English needs to be revised to reach the language standards that are required by the journal. Furthermore, all typing errors should be corrected.

References 27-28 should be corrected.

Author Response

Reviewer #3: In my opinion the paper merits publication in Molecules.

  1. In some sentences English needs to be revised to reach the language standards that are required by the journal. Furthermore, all typing errors should be corrected.

R: Thank you very much for your recognition of our article, which gives us great encouragement. Problems with grammar and spelling errors in the manuscript have been corrected.

  1. References 27-28 should be corrected.

R: Thank you very much for your kindly reminding. The errors in the references in the manuscript have been corrected.

Round 2

Reviewer 2 Report

Within the revised work, the authors have introduced rather minor changes to the manuscript and most of the comments/suggestions have not been incorporated but rather deemed from their side unnecessary. Nevertheless, some parts of the work have been edited and thus the entire work reads to a certain extent better.